# Principle and Applications of Multimode Strong Coupling Based on Surface Plasmons

**DOI:** 10.3390/nano12081242

**Published:** 2022-04-07

**Authors:** Zhicong He, Cheng Xu, Wenhao He, Jinhu He, Yunpeng Zhou, Fang Li

**Affiliations:** 1School of Mechanical and Electrical Engineering, Hubei Key Laboratory of Optical Information and Pattern Recognition, Wuhan Institute of Technology, Wuhan 430073, China; hzc_900503987@163.com (Z.H.); xc578063977@163.com (C.X.); hwh99970711@163.com (W.H.); hejinhu1998@163.com (J.H.); z1433845556@163.com (Y.Z.); 2School of Mechanical and Electrical Engineering, Hubei Polytechnic University, Huangshi 435003, China; 3Hubei Key Laboratory of Intelligent Transportation Technology and Device, Hubei Polytechnic University, Huangshi 435003, China

**Keywords:** surface plasmon, interaction, multimode strong coupling

## Abstract

In the past decade, strong coupling between light and matter has transitioned from a theoretical idea to an experimental reality. This represents a new field of quantum light–matter interaction, which makes the coupling strength comparable to the transition frequencies in the system. In addition, the achievement of multimode strong coupling has led to such applications as quantum information processing, lasers, and quantum sensors. This paper introduces the theoretical principle of multimode strong coupling based on surface plasmons and reviews the research related to the multimode interactions between light and matter. Perspectives on the future development of plasmonic multimode coupling are also discussed.

## 1. Introduction

Surface plasmon polaritons [1,2,3] (SPPs) are localized electromagnetic wave generated by the collective oscillation of electrons on metal surfaces, that are decayed exponentially in the direction perpendicular to the interface of both media [4]. It has become a popular research topic in the field of nanophotonics in recent years. Owing to their distinctive optical properties [5,6,7,8,9], SPPs can break through the optical diffraction limit and strengthen the near-field enhancement effect, providing an opportunity to explore the interaction between light and matter in micro/nano fields.

In the last few decades, owing to the potential applications of SPPs and excitons in sensors, light emitters and nano-optical devices, considerable efforts have been devoted to the coupling. SPPs represent a basic and interesting multimode coupling system. It is greatly affected by the structural parameters and can be adjusted independently. Strong coupling between SPPs and organic molecules have always been the subject of intense investigation [10] Essentially, SPPs can be considered as resonant–element–excited states [11,12,13], which are formed by the coherent coupling of quasi-free electron gas and the collective oscillation along the metal surface after the interaction between light and matter. Particularly, a new resonance, surface plasmon resonance (SPR), may be generated, whereas the resonance frequency in the metal SPPs is equal to the incident light [14,15,16]. It is worth noting that the frequency is affected by the size and morphology of the metal nanoparticles (NPs). On this basis, a variety of new composite structures can be produced by combining the metal structure with the material molecules. The study of the interaction between SPPs and molecules is of great significance not only in basic physics, but also in the research and development of various new nano-optoelectronic devices, as well as the development of quantum communication [17].

Two different regimes are divided which are according to the coupling of photon and exciton states, namely weak coupling (WC) and strong coupling (SC). In WC [18,19,20,21], there is no disturbance between the wave functions, and the energy exchange rate between the photon and exciton changes, and the interaction strength is less than the damping [22], whereas the energy levels of the SPPs and exciton modes are constant, and the spatial and spectral distributions are altered while the exciton dynamics are changed only mildly [23]. The main function of plasmons is to induce the molecules to enhance the absorption fluorescence. However, in SC [24], mixed states may lead to strongly modified excitonic dynamics. It produces the formation of half-light/half-matter quasiparticles termed polaritons. The terms of enhancing the Raman scattering process [24,25,26] are investigated in the polariton states formed by the coupling between excitons and the cavity. When the Raman excitation energy becomes resonant with polariton states, the maximum intensity can be observed. An abundance of results related to Raman scattering have been achieved by Professor Mlayah’s team [27,28,29,30] from research conducted at the Centre d’Elaboration des Matériaux et d’Etudes Structurales (CEMES) facility located in Toulouse, France. In addition, SC is manifested as the anticrossing of the coupling modes and as the resonance of two equal-intensity transitions, which are separated as vacuum Rabi splitting [31,32,33,34], while the corresponding frequency of the molecular energy level resonates with the starting frequency of the SP exciton. A mixture of cavity photon modes and optical transitions is considered as a new cavity polaronic mode, in which the cavity photon lifetime will affect the radiative lifetime. When the interaction is strong enough, the energy levels change due to perturbations between the wave functions [35,36,37,38,39].

Recently, multimode coupling, in which the excitonic modes are coupled to multiple photonic modes, has attracted increasing attention. A common method to achieve multimode coupling is to introduce coupled cavities [40,41], or a planar cavity containing multiple cavity modes [42] with an extended optical path length. Another method is to construct nanostructures which can support varied plasmonic modes or sustain both plasmonic and cavity modes [43]. Compared with the classic single-mode strong coupling system [44,45,46,47], the multimode strong coupling system represents a more meaningful direction in the field of plasmon–molecule interactions. Compared with single coupling systems [48,49], multimode strong coupling systems have more energy dissipation channels and larger modulation. This hybrid system possesses a mixed plexciton state with diverse excitons, which supplies an underlying technical approach to actively control the strong exciton–plasmon–exciton coupling through modulating distinct excitons. The development of multimode strong coupling systems has led to an efficient super-fast energy transfer pathway between both exciton states. Therefore, the study of multimode strong coupling will open a host of possibilities for the development and improvement of strongly coupling plasmon nanostructures. The occurrence of multimode Rabi splitting in such systems is of significance and relevance in both basic and applied sciences, providing a promising opportunity for multimode hybridization and energy transfer. In addition, multimode strong couplings also possess unlimited prospects in the field of multiple entanglement and the realization of quantum computing devices using multiple quasiparticle boson systems. Furthermore, the nanomanipulations of plasmonizing nanoobjects will open avenues for the design of structures allowing or prohibiting plasmon/exciton couplings [50,51,52].

This study opens new possibilities for exploring and utilizing the unique properties of hybrid states based on SPPs technology. Starting from the basic principles, we focus on the recent developments in multimode coupling technology. Moreover, the future trends of multimode strong coupling between light and matter are discussed.

## 2. Basic Principle of Multimode Strong Coupling

### 2.1. Vacuum Rabi Splitting

The Hamiltonian was proposed by Dicke [53], giving a model that is essential and crucial in quantum optics, which describes systems relating to the interaction of light and materials, and it represents the simplest model. In addition, the Dicke Hamiltonian model [54,55,56] involves the collection interaction of N two-level atoms with energy separation equal to $\hbar {\tilde{\omega }}_{A}$ with a one-mode radiation field of frequency ${\tilde{\omega }}_{F}$. It derived the following assumptions from the quantization of electromagnetic fields: (i) The dipole approximation, or the long wavelength limit, i.e., the electric field is calculated at the center of mass of the atoms and is independent of position. (ii) Only two atomic energy levels will interact with electromagnetic fields. The lowest energy state is at least metastable, and the decay to other energy levels is negligible. The Dicke Hamiltonian model for N identical two-level atoms immersed in a one-mode electromagnetic field is as follows [57]:(1)$$\hat{H}={\hat{a}}^{†}\hat{a}+{\tilde{\omega }}_{A}{\hat{J}}_{z}+\frac{g}{\sqrt{N}}({\hat{a}}^{†}+\hat{a})({\hat{J}}_{+}+{\hat{J}}_{-})$$
where $\omega_{A}={\tilde{\omega }}_{A}/{\tilde{\omega }}_{F}\geq 0$ is given in units of the frequency of the field, and $g=\tilde{g}/{\tilde{\omega }}_{F}$ is the (adimensional) coupling parameter. In addition, the operators $\hat{a}$ and ${\hat{a}}^{†}$ are the one-mode annihilation and creation photon operators, respectively, ${\hat{J}}_{z}$, ${\hat{J}}_{\pm }$ are the atomic relative population operator and atomic transition operators. Furthermore, the Dicke Hamiltonian model shows wide applications in many fields [58,59,60,61,62].

The famous Jaynes–Cummings (JC) model [63,64,65] is the simplest but nontrivial model for depicting the interaction of a two-level atom with an electromagnetic field at resonance, describing the radiation field and rotating wave approximation, additionally, it can be usually described as the most common soluble quantum-mechanical model of a single atom in an electromagnetic field, it represents the single-excitation sub-space of the full Jaynes–Cummings model:(2)$$\overset{\wedge }{H}=\hbar \omega_{pl}{\hat{a}}^{†}\hat{a}+\hbar \omega_{em}{\hat{\sigma }}^{†}\hat{\sigma }+\hbar g({\hat{a}}^{†}\hat{\sigma }+{\hat{\sigma }}^{†}\hat{a})$$
where $\hat{a}$ and ${\hat{a}}^{†}$ are also operators, which are the same as Equation (1), and $\omega_{pl}$ represents the plasmon resonance frequency, while $\omega_{em}$ is the emitter transition frequency, in addition, $\hat{\sigma }$ is the Pauli matrices for the two-level system. The decoherence and damping of the emitter and plasmon are neglected in this condition.

The coupling strength between plasmon and emitter, *g* is as follows:(3)$$g=\frac{2{\vec{\mu }}_{em}\cdot {\vec{\varepsilon }}_{0}}{h}$$
where the transition dipole matrix element of emitter is ${\vec{\mu }}_{em}$ and the local electric field generated by the plasmon at the position of the emitter is ${\vec{\varepsilon }}_{0}$. For the electromagnetic fields which were confined into a cavity of mode volume *V*, the maximum *g* (for the emitter is at the maximum value of the field, whose dipole moment corresponds to the polarized direction of the field) is:(4)$$g=\frac{\mu_{em}}{h}\sqrt{\frac{h\omega_{pl}}{2V}}$$

Diagonalization of the Hamiltonian in Equation (2), the classical coupled harmonic oscillator model is often used to predict the upper polariton branches (UPB) and lower polariton branches (LPB) [66,67,68], it provides the mode frequencies of the polaritons:(5)$$\omega_{\pm }=\frac{1}{2}(\omega_{pl}+\omega_{em})\pm \Omega_{R}$$
where the vacuum Rabi frequency $\Omega_{R}$ is
(6)$$\Omega_{R}=\sqrt{g^{2}+1/4{\left(\omega_{pl}-\omega_{em}\right)}^{2}}$$
$\Omega_{R}$ represents the frequency at which the energy oscillates between the plasmon and the emitter.

Compared with the intrinsic linewidths of optical modes and excitons, Rabi splitting [69,70,71] is larger and more capable of maintaining a strong coupling state, which may lead to two independent modes: UPB and LPB, as shown in Figure 1a. While the two independent states are degenerated and owed equal lifetimes and effective masses, the strong coupling, which demonstrated anti-crossing and the appearance of two transitions of equal strength separated by Rabi splitting, the two new modes produced in the strong coupling state can be explained through mode hybridization, analogous to the molecular orbital theory [72,73]. In addition, the strong coupling regime illustrated by two coupled oscillators can be seen in Figure 1b [74], this system composed exciton A and plasmon. An important consequence of strong coupling is that the exciton should possess the plasmonic properties due to the coherent interactions, such as polarization and emission direction [75,76,77], and the exciton–plasmon coupling, including enhanced cooperative emission and enhanced exciton transport, in the excited state molecules and molecular assemblies interacting with optical modes localized in cavities and other nanostructures. A significant limitation of the single molecules strong-coupling is that the operation may be required under cryogenic temperature [78], and that the presence of nonlocality imposes, will limit strictly the practical applications [79].

Furthermore, damping might be taken into consideration through the Heisenberg–Langevin method, in which both the plasmon and emitter are coupled to a library of harmonic-oscillator modes [80]. The Green’s function approach [81] can be used to calculate the absorption spectra, while the Liouville approach [82] can be used to depict the system dynamics. Either the Liouville equations or the Heisenberg–Langevin [83,84] may both calculate the absorption and scattering cross-sections, through introducing the external electric field, ε, that acts on the plasmon:(7)$${\hat{H}}_{d}=-\varepsilon {\hat{\mu }}_{pl}$$
where ${\hat{\mu }}_{pl}=d_{pl}(\hat{a}+{\hat{a}}^{†})$ is the plasmon dipole operator and $d_{pl}$ is the plasmon dipole moment. Owing to the interaction between the emitter and light being much weaker than that of the plasmonic nanostructure, the direct driving of the emitter by the external field can be ignored. Then, the optical response can be reached by solving the equations of motion in frequency or time domains, respectively.

Several assumptions are involved in this method of analysis. By regarding the emitter as a two-level system, the complex energy level structure that exists in any quantum emitter is ignored. By regarding the plasmon field as a population of a single (quasi-) normal modes, additional modes that may overlap in frequency and any frequency dependence of the plasmon damping rate are ignored. Finally, it regards the emitter as a point dipole, not taking into account any effects due to magnetic field gradients in the spatial range of the emitter. Although the model is simplified, it can still process the experimental data quantitatively.

The solution of the quantum model requires large computational resources for accounting for all the potentially occupied plasmon substates. If the plasmon field can be approximated as the classical one, that is, significant acceleration can be obtained while the effect of quantum fluctuations in the field can be ignored, then in this condition, the Heisenberg–Langevin or Liouville equations may be reduced as the Maxwell–Bloch [85,86,87], which is semiclassical. Matthew’s team [78] have verified that the predictions of the quantum model and the semiclassical model are almost identical in the case of parameters related to plasmon coupling, the Maxwell Bloch equation is as follows [88]:(8)$${\ddot{\mu }}_{pl}+\gamma_{pl}{\dot{\mu }}_{pl}+\omega_{pl}^{2}\mu_{pl}=F_{0}+g(\omega_{pl}d_{pl}/d_{em})\mu_{em}$$
(9)$$\{\begin{array}{c} {\dot{\rho }}_{1}^{em}=\omega_{em}\rho_{2}^{em}-\gamma_{2}\rho_{1}^{em}\\ {\dot{\rho }}_{2}^{em}=-\omega_{em}\rho_{1}^{em}-(g/d_{pl})\mu_{pl}\rho_{3}^{em}-\gamma_{2}\rho_{2}^{em}\\ {\dot{\rho }}_{3}^{em}=(g/d_{pl})\mu_{pl}\rho_{2}^{em}-\gamma_{1}(\rho_{3}^{em}+1) \end{array}$$
where $d_{em}$ represents the emitter’s transition dipole moment, $\gamma_{pl}$ is the plasmon decay rate, $\gamma_{1}$ is the emitter’s energy decay rate, $\gamma_{2}$ is the emitter’s dephasing rate, $\rho_{1}^{em}=2ℜ[\rho_{01}]$, $\rho_{2}^{em}=-2ℑ[\rho_{01}]$, $\rho_{3}^{em}=\rho_{11}-\rho_{00}$, and $F_{0}=4\omega_{pl}d_{pl}^{2}\varepsilon$. The pure dephasing of the plasmon can be ignored for the slowing energy damping. Differentiation of the first of Equation (9) with respect to time and substitution of the second equation into the first obtains ${\ddot{\rho }}_{1}^{em}=\omega_{em}[-\omega_{em}\rho_{1}^{em}-(g/d_{pl})\mu_{pl}\rho_{3}^{em}-\gamma_{2}\rho_{2}^{em}]-\gamma_{2}{\dot{\rho }}_{1}^{em}$. In the case of the linear response (scattering or absorption) of the coupled system, further approximations regarding the coherence between the emitter ground state and the excited state can also be ignored, and the emitter’s population almost remains in the ground state; For example, $\rho_{3}^{em}\approx -1$ and $\rho_{2}^{em}\ll 1$. Using $\mu_{em}=d_{em}\rho_{1}^{em}$, we obtain:(10)$${\ddot{\mu }}_{em}+\gamma_{em}{\dot{\mu }}_{em}+\omega_{em}^{2}\mu_{em}=g(\omega_{em}d_{em}/d_{pl})\mu_{pl}$$

Assuming the case where the dephasing of the emitter is much faster than energy decay, the total emitter decay rate is $\gamma_{em}\approx \gamma_{2}$. This assumption is suitable for all experiments at or near room temperature. Equations (8) and (10) describe a pair of coupled classical harmonic oscillators. The displacement of each harmonic oscillator corresponds to the dipole moments of the plasmon and exciton, respectively. Generally, the electromagnetic near field couples via two oscillators. The plasmon oscillator is driven only by an external field. An intuitive and simple picture of this coupling exciton–emitter system is provided by the coupled oscillator model, and the oscillation of two coupled pendulums is easily solved, which can be found in reference [74,89].

In the absence of the external driving field, the coupled oscillator equation can be solved easily in the frequency domain to obtain the normal modal frequency of the system. If assuming the coupling strength, damping rate and detuning are smaller than resonance frequency ($g,\gamma_{em},\gamma_{pl},\left|\omega_{em}-\omega_{pl}\right|\ll \omega_{pl}$), and the normal-mode frequencies are:(11)$$\omega_{\pm }=\frac{1}{2}(\omega_{pl}+\omega_{em})-\frac{1}{4}i(\gamma_{pl}+\gamma_{em})\pm \Omega_{R}$$
where now
(12)$$\Omega_{R}=\sqrt{g^{2}+1/4{\left(\omega_{pl}-\omega_{em}\right)}^{2}-1/16{\left(\gamma_{pl}-\gamma_{em}\right)}^{2}}$$

$\gamma_{pl}$ and $\gamma_{em}$ are the uncoupled SPPs and exciton decay rates, respectively. Comparison with Equations (5) and (6), shows that the results of the coupled-oscillator model are the same as the Jaynes–Cummings Hamiltonian under the constraints of zero damping and dephasing, if $\Omega_{R}$ is real while the normal modes are non-degenerate. In the cases of $\left|\omega_{em}-\omega_{pl}\right|\ll \omega_{pl}$, and $\gamma_{em}<\gamma_{pl}$ (in all experiments that have been shown, both were fulfilled), this case reduces to
(13)$$g>\frac{1}{4}(\gamma_{pl}-\gamma_{em})$$

Between over-damped and the under-damped cases, the transition can be expressed as a boundary between weak and strong coupling. However, strong coupling is considered to occur only when at least one complete Rabi oscillation occurs [90], when:(14)$$g>\frac{1}{4}(\gamma_{pl}+\gamma_{em})$$

Generally, many researchers use Equation (11) to judge whether the system is in the strong-coupling state or not. However, it is significant to realize that it does not represent a sharp threshold, when coupling plasmon-emitter systems cross this boundary, their behavior does not change qualitatively.

Moreover, the coupled oscillator equation can be easily generalized to *N* emitters independently coupled to a single plasmon mode, each with a different coupling constant $g_{i}$. In the condition, the strong-coupling leads to
(15)$$\sqrt{N}g_{rms}>\frac{1}{4}(\gamma_{pl}+\gamma_{em})$$
where $g_{rms}=(1/N)\sum_{i=1}^{N}g_{i}$ and where assuming the same linewidth, $\gamma_{em}$, for all emitters.

Up until now, the JC model proposed by Tavis and Cummings and its extension, the Tavis–Cummings (TC) model, are still the basis of studying the basic properties of quantum electrodynamics (QED) and understanding the existence of Rabi oscillations. To our knowledge, the SPPs–exciton strong coupling system is related to harmonic oscillators, and thus the definition of strong coupling is dependent on, among other things, the conventions of the particular field of physic involved; the system is in the strong coupling regime whenever the vacuum Rabi splitting (often called Rabi splitting) is experimentally observable [91].

### 2.2. Multiple Harmonic Oscillators

Furthermore, the multimode system is similar with multiple harmonic oscillators, considering the coupled three-harmonic-oscillator system as an example, as shown in Figure 2a. When coupling occurs, the frequencies of the oscillators will change, which is related to the coupling strength. Similarly, as shown in Figure 2b, when SPPs–double exciton coupling occurs, the plasmon energy level $U_{pl}$, exciton A energy level $U_{ea}$, and exciton B energy level $U_{eb}$ will change, and hybridization will occur, producing three hybridized plexciton states and double Rabi splitting. After the coupling, there are three unique solutions for U ($U_{H}$, $U_{M}$, and $U_{L}$), and thus the polariton energy dispersion comprises three energy branches. These are the aforementioned UPB and LPB, along with middle polariton branches (MPB). This change is evident in the scattering, absorption, and extinction spectra of the coupled system, with three peaks and two dips. When the plasmon resonance energy is coupled from low to high with excitons of constant energy, a double anticrossing image occurs, which is a typical characteristic of the multimode strong coupling between two excitons (excitons A and exciton B) and a plasmon, as illustrated in Figure 2c.

To provide physical insights into plasmon–exciton multimode coupling, we further focus on the linear analysis of the scattering spectra, using the coupled oscillator model to fit the calculation results [93,94]. In the hybrid system, the coupling oscillators represent the plasmon, exciton A, and exciton B. The motion equations for the three oscillators are as follows [95,96]:(16)$$\overset{\cdot \cdot }{x_{pl}(t)}+\gamma_{pl}\overset{\cdot }{x_{pl}(t)}+\omega_{pl}^{2}x_{pl}(t)+g_{a}\overset{\cdot }{x_{a}}(t)+g_{b}\overset{\cdot }{x_{b}}(t)=F_{pl}(t)$$
(17)$$\overset{\cdot \cdot }{x_{a}(t)}+\gamma_{a}\overset{\cdot }{x_{a}(t)}+\omega_{a}^{2}x_{a}(t)-g_{a}\overset{\cdot }{x_{pl}}(t)=F_{a}(t)$$
(18)$$\overset{\cdot \cdot }{x_{b}(t)}+\gamma_{b}\overset{\cdot }{x_{b}(t)}+\omega_{b}^{2}x_{b}(t)-g_{b}\overset{\cdot }{x_{pl}}(t)=F_{b}(t)$$
where, $x_{pl}$, $x_{a}$ and $x_{b}$ represent plasmon, exciton A oscillator, and exciton B oscillator, respectively. $\gamma_{pl}$, $\gamma_{a}$ and $\gamma_{b}$ are the damping rate of plasmon, exciton A and exciton B, respectively, $\omega_{pl}$, $\omega_{a}$ and $\omega_{b}$ are the resonance frequencies of plasmon, exciton A and exciton B, respectively, $g_{a}$ is the coupling rate between the plasmon and exciton A, while $g_{b}$ is the coupling rate between the plasmon and exciton B, $F_{pl}$, $F_{a}$ and $F_{b}$ represent the driving forces due to the external source. We assumed that both excitons A and B were both entirely driven by the plasmon oscillator; hence, we set $F_{pl}(t)=F_{pl}e^{-i\omega t}$, where ω is the frequency of the electric field, $F_{a}(t)=0$, $F_{b}(t)=0$. Finally, $x_{pl}(t)$, $x_{a}(t)$ and $x_{b}(t)$ can be derived from Equations (16)–(18). In some specific hybrid nanostructures, the dimensions of the structures are small compared to the optical wavelength; thus, the scattering cross-section can be calculated in the quasi-limit [97]. In this limitation, the scattering cross-section is $(8\pi /3)\cdot k^{4}{\left|F_{pl}x_{pl}\right|}^{2}$, where k=ωn/c is the wave vector of light. By substituting $x_{pl}(t)$ into $(8\pi /3)\cdot k^{4}{\left|F_{pl}x_{pl}\right|}^{2}$ and the use of the incident light energy E replaces the incident light frequency ω, the scattering cross-section can be obtained as follows:(19)$$\sigma_{scat}(E)=(8\pi /3)\cdot k^{4}{\left|F_{pl}x_{pl}\right|}^{2}\infty E^{4}{\left|\frac{ab}{ab(E^{2}-E_{pl}^{2}+iE\gamma_{pl})-E^{2}g_{a}^{2}b-E^{2}g_{b}^{2}a}\right|}^{2}$$
where $a=E^{2}-E_{a}^{2}+iE\gamma_{a}a$, and $b=E^{2}-E_{b}^{2}+iE\gamma_{b}b$.

The multimode coupling system can be explored by description of a three-coupled oscillator model composed of the optical modes [98,99], and the interaction Hamiltonian for the system can be expressed as [100]:(20)$$\left[\begin{array}{ccc} U_{pl}-i\frac{\gamma_{pl}}{2} & g_{b} & g_{a}\\ g_{b} & U_{b}-i\frac{\gamma_{b}}{2} & 0\\ g_{a} & 0 & U_{a}-i\frac{\gamma_{a}}{2} \end{array}\right]\left[\begin{array}{c} \alpha_{pl}\\ \alpha_{b}\\ \alpha_{a} \end{array}\right]=U\left[\begin{array}{c} \alpha_{pl}\\ \alpha_{b}\\ \alpha_{a} \end{array}\right]$$
where $U_{pl}-i\frac{\gamma_{pl}}{2}$ and $U_{b}-i\frac{\gamma_{b}}{2}$ are the undisturbed energies of the optical modes in excitons A and B, respectively, $U_{a}-i\frac{\gamma_{a}}{2}$ is the exciton energy, $g_{b}$ is the coupling constant between optical modes, and $g_{a}$ is the exciton–photon coupling potential. $\alpha_{pl}$, $\alpha_{b}$, and $\alpha_{a}$ are the eigenvector components (Hopfield coefficients), the corresponding ${\left|\alpha_{pl}\right|}^{2}$,${\left|a_{b}\right|}^{2}$, and ${\left|\alpha_{a}\right|}^{2}$ represent the weighting efficiencies and satisfy ${\left|\alpha_{pl}\right|}^{2}+{\left|\alpha_{b}\right|}^{2}+{\left|\alpha_{a}\right|}^{2}=1$. Solution of the long-term determinant of resonance where $U_{pl}-i\frac{\gamma_{pl}}{2}=U_{b}-i\frac{\gamma_{b}}{2}=U_{a}-i\frac{\gamma_{a}}{2}$ (all of them are zero) leads to the eigenvalues $U=0,\pm \sqrt{\left(g_{a}^{2}+g_{b}^{2}\right)}$. Thus, it is expected that there might exist three modes, namely the central mode under the unperturbed energy and the external two modes shifted by $\pm \sqrt{\left(g_{a}^{2}+g_{b}^{2}\right)}$ to a higher and lower energy. The eigenvectors Ψ of the UPB, MPB and LPB modes are [100,101,102]:(21)$$\Psi_{UPB}=\frac{1}{\sqrt{2}}\phi_{pl}+\frac{\Delta }{\sqrt{2}{\left(\Omega^{2}+\Delta^{2}\right)}^{1/2}}\phi_{b}+\frac{\Delta }{\sqrt{2}{\left(\Omega^{2}+\Delta^{2}\right)}^{1/2}}\phi_{a}$$
(22)$$\Psi_{{}_{MPB}}=\frac{\Delta }{\sqrt{2}{\left(\Omega^{2}+\Delta^{2}\right)}^{1/2}}\phi_{b}-\frac{\Delta }{\sqrt{2}{\left(\Omega^{2}+\Delta^{2}\right)}^{1/2}}\phi_{a}$$
(23)$$\Psi_{{}_{LPB}}=\frac{1}{\sqrt{2}}\phi_{pl}-\frac{\Delta }{\sqrt{2}{\left(\Omega^{2}+\Delta^{2}\right)}^{1/2}}\phi_{b}-\frac{\Omega }{\sqrt{2}{\left(\Omega^{2}+\Delta^{2}\right)}^{1/2}}\phi_{a}$$
where $\phi_{pl}$, $\phi_{b}$, and $\phi_{a}$ are the unperturbed basis functions. With Equation (22), no component from the basis function $\phi_{pl}$ is contained by the eigenvector of the central mode, of exciton A; for the central mode, the amplitude of the light field in the cavity containing this condition is zero. All weights of $\phi_{pl}$ reside equally in the two external components. In comparison, if light is entering into the empty cavity, then the central mode will be bright (for the nonzero $\phi_{d}$ component in Equation (22)).

Surprisingly, the nature of the (3 × 3) matrix Hamiltonian of Equation (20) is shown, which has one basis function ($\phi_{pl}$) that is coupled to the other two basic functions $\left(\phi_{b},\phi_{a}\right)$. $\phi_{b}$ and $\phi_{a}$ by contrast are not coupled directly. The eigenvector $\Psi_{MPB}$ is the central mode. At the same time, these two external modes contain components from both $\phi_{pl}$ and $\phi_{b}$, with thus observation in both directions.

This situation for the coupled two-cavity one-exciton case is similar to that found for the normal modes of three masses connected linearly by springs [103] or of linear triatomic molecules [104]. For which cases, there is one mode corresponding to the zero displacement of the central mass, and the two outer masses undergo tensile mode vibrations. This is consistent with Equation (22) where in the central mode the weight of the oscillator ($\phi_{pl}$) connected with the other two is zero.

### 2.3. Scattering Spectra

For analyzing the coupling strength in the plasmonic system, which cannot be obtained directly from Fano’s original spectrum, the coupled harmonic oscillator model served as a simulation of the scattering spectrum for describing the real physical situation, phenomenologically [95]. Thus, fitting of the calculated spectra into the coupled-oscillator model, which makes it possible to determine whether the system is in a strong-coupling regime or not, was performed for determining which phenomenon (hybridization or interference) is responsible for the spectra.

The coupled-oscillator equations can also be solved with a driving force to obtain the absorption and scattering cross sections. When the frequency of driving force is ω under the steady state, the solution is:(24)$$\mu_{pl}=\frac{F_{0}(\omega_{em}^{2}-\omega^{2}-i\omega \gamma_{em})}{(\omega_{em}^{2}-\omega^{2}-i\omega \gamma_{em})(\omega_{pl}^{2}-\omega^{2}-i\omega \gamma_{pl})-\omega_{em}\omega_{pl}g^{2}}$$

The scattering cross-section is given by [95]:(25)$$\sigma_{scat}(\omega )\infty \omega^{4}{\left|\mu_{pl}\right|}^{2}$$

Similar conclusions can be obtained for absorption and extinction cross-sections [95]. From Equations (24) and (25), the calculated spectra indicate unique plasmon and exciton parameters and different coupling strengths.

In the previous literature [95,105], the following represent various coupled-oscillator equations with different coupling terms:(26)$$\{\begin{array}{c} {\ddot{\mu }}_{pl}+\gamma_{pl}{\dot{\mu }}_{pl}+\omega_{pl}^{2}\mu_{pl}=F_{0}-g{\dot{\mu }}_{em}\\ {\ddot{\mu }}_{em}+\gamma_{em}{\dot{\mu }}_{em}+\omega_{em}^{2}\mu_{em}=g{\dot{\mu }}_{pl} \end{array}$$

Generally, the coupling terms used in Equations (8) and (10) are preferred, since the equation for the leading choice of the coupling term is derived from a quantum mechanical model in the linear and classical limitations. However, if the coupling term in Equation (26) is used, the steady-state solution is different only in the last term of the denominator:(27)$$\mu_{pl}=\frac{F_{0}(\omega_{em}^{2}-\omega^{2}-i\omega \gamma_{em})}{(\omega_{em}^{2}-\omega^{2}-i\omega \gamma_{em})(\omega_{em}^{2}-\omega^{2}-i\omega \gamma_{em})-\omega^{2}g^{2}}$$

This will give almost the same results as Equation (24), while $g,\gamma_{em},\gamma_{pl}\ll \omega$; this has been the case for all reported experimental results [95,106,107].

Furthermore, a coupled harmonic oscillator model elucidates the nature of the interaction and reveals the coherent quantum superposition of the excitons, which are mediated by the plasmonic interaction. In addition, each component of the coupled system is described as a harmonic oscillator with its own resonance damping and frequency.

The theory discussed here is based on the characteristics of SPPs when coupled with excitons, which are confined to the metal surface at the nanometer scale, thus greatly compressing the spatial distribution of the electromagnetic field [108,109]. Furthermore, this spatial variation will introduce substantial theoretical complexities in the coupled oscillator models [77]. This forms a theoretical basis for exploring the coupling effect, clearly, the coupling and hybridization phenomena open an avenue for the enhancement of plasmonic resonances in nanostructures, which deserves further study.

## 3. Recent Progress in Multimode Strong Coupling

### 3.1. Multimode Coupling Related to Microcavity Nanostructure

#### 3.1.1. Coupling Related to TMDs Microcavity Nanostructure

Two-dimensional transition metal dichalcogenides (TMDs) have recently attracted widespread attention owing to their unique electrical and optical properties. The TMDs display a strong enhancement of the Coulomb interaction [110]. The resulting bound electron-hole pairs, govern the electrical charge and optical transport properties, then affect the Coulomb interaction between the oscillator strength and exciton binding energy [111]. The exciton absorption of TMDs is located within the visible light band, showing broader application prospects and attracting the interest of researchers. TMDs have opened a new research challenge for strong coupling. Menon M. et al. [112] demonstrated the strong coupling between two-dimensional excitons and cavity photons in a monolayer MoS_2_-based microcavity with a Rabi splitting of 46 ± 3 meV for the first time. The same team [113] presented an approach for dynamically regulating the interaction between excitons in monolayer WS2 and microcavity photons at room temperature, and observed a Rabi splitting of 60 meV.

Furthermore, TMDs monolayers provide the opportunity to study multimode strong coupling with larger Rabi splitting, and temperature might be a key factor affecting the multiple coupling. Cuadra J. et al. [114] reported the strong interaction between localized surface plasmon resonance (LSPR) in silver nanoprisms and excitons and trions in monolayer tungsten disulfide (WS_2_). The high density of the photonic states is shown in Figure 3a at the corners of the nanoprism, which overlaps with the WS_2_ monolayer for efficient plasmon−exciton interaction, and the inset shows the SEM image of such a particle and a magnified view of a corresponding dark-field image. In addition, for T = 300 K, the Rabi splitting was 120 meV, whereas in a low temperature case, T = 6 K (Figure 3b, red, green, and blue solid lines represent the eigen energies extracted from the Hamiltonian analysis. Black solid lines indicate exciton, trion and plasmon resonances), three anticrossing bands, corresponding to UPB, MPB and LPB, respectively, with minimal splitting measured approximately 150 meV. The 30 meV increase can be explained by the detuned exciton and trion. Both plasmon and exciton resonances are narrowed upon cooling, and exciton line narrowing at low temperatures is consistent with previous results [115]. Figure 3c shows the Hopfield coefficients for the plasmon, exciton, and trion contributions to the UPB, MPB, and LPB states as a function of the plasmon resonance. Double Rabi splitting was observed in the dark-field scattering spectra using a plasmonic nanostructure interacting with two types of excitons (charged and neutral) in WS_2_ with decreasing temperature. The degree of plasmon–exciton–trion coupling might be tailored by varying the temperature.

Moreover, similar exploration was conducted by other groups, and the size of the nanosystem was also related to the strength of the multimode coupling. Li B. et al. [116] demonstrated a large 300 meV Rabi splitting that was achieved under ambient conditions in a strong coupling regime by Ag–WS_2_ heterostructure embedding. The system consisted of a 100 nm Ag mirror and a 30 nm MgF_2_ spacer with the Ag nanodisks fabricated by e-beam lithography (EBL), as shown in Figure 4a. In the plasmon–exciton-cavity, the 99 nm Ag nanodisk was the best structure for the strongest interaction, with a 300 meV Rabi splitting among the three oscillators, which can be seen in Figure 4b. Red dots with error bars show energies obtained from the reflection spectrum. The horizontal black dashed lines represent the A-exciton and the microcavity resonant energy, respectively. The black slanted short-dashed line represents plasmon resonance mode. It depicts resulting dispersion curves of the Ag–WS_2_ heterostructure that was embedded in the optical microcavity. They proposed the criteria of the strong coupling for the three oscillators, and the Hopfield coefficients of the three branches were calculated as shown in Figure 4c. Each branch consisted of part-plasmon, part-exciton, and part-cavity modes, demonstrating that the hybrid state was successfully generated by combining the Ag–WS_2_ heterostructure with the optical microcavity. At a nanodisk diameter of 99 nm, the strongest interaction of 300 meV Rabi splitting was obtained. Thus, it is clearly evident that the size of the Ag nanodisk affects the strong coupling.

Jiang P. et al. [117] proposed a strong exciton–plasmon–exciton coupling system that consisted of a silver nanoprism separated from a monolayer WS_2_ by J-aggregates, as depicted in Figure 5a, which was performed by finite-difference time-domain (FDTD) simulations to obtain the optical response. Figure 5b depicts the scattering cross-section spectra of the hybrid nanostructure. The solid white lines indicate the three anticrossing bands corresponding to the UPB, MPB and LPB, and the solid white lines match well with the numerical simulation results; the splitting extracted between the UPB and MPB is 130 meV at the zero detuning while the splitting between the MPB and LPB is 170 meV. Since the damping losses were not considered, the calculated Rabi splittings are larger than the actual results slightly. Furthermore, Figure 5c shows the weighting efficiencies for the LSPR mode, exciton A of WS_2_, and J-aggregates exciton contributions to three polariton branches as a function of the plasmon resonance.

Overall, the research interest in TMDs mainly focuses on their distinctive optical properties: strong nonlinear optics response, effective valley-spin coupling, strong exciton binding energy, large spin–orbit interaction, and transition dipole momentum of the material. These properties of multimode strong coupling systems based on TMDs and their related characteristics are now sought after by researchers. Such a system has potential applications in optical modulators at the nanoscale and polaritonic devices based on ultrathin materials.

#### 3.1.2. Coupling Related to Single-Dye Microcavity Nanostructure

Although most studies about strong coupling have been explored interactions between a single excitonic mode and a single photonic mode, some teams have focused on multimode mixed coupling, in which the excitonic mode is coupled to more than one photonic mode; however, these systems are complicated, and the quality of excitons are more or less affected.

Zhang K. et al. [118] presented a hybrid strong coupling between multiple photonic modes and excitons in an organic-dye-attached photonic quasicrystal. The electron beam evaporation method was used to deposit the dielectric multilayers, and TDBC dye was selected for the J-aggregates, acting as an organic semiconductor. Following the simulation parameters, SiO_2_/Ta_2_O_5_ multilayers were deposited onto K9 glass substrate, and then the J-aggregates were directly spun onto the top surface, as schematically described in Figure 6a. The value of the coupling energy, namely the coupling energy between the photonic mode and the excitons, was found to be $h\Omega_{H}=67\;\mathrm{meV},h\Omega_{C}=93\;\mathrm{meV}$ and $h\Omega_{L}=85\;\mathrm{meV}$. The six polariton bands of the calculated eigen energies in this system are described by the dashed lines in Figure 6b, and all these results fit well with the experimental data.

Balasubrahmaniyam M. et al. [119] investigated and experimentally implemented the photonic analog of localization induced by ultra-strong interactions in a coupled three-mode system. Furthermore, they demonstrated the ultra-strong coupling between a highly dispersive cavity plasmon mode and dimer excitons of J-aggregates Rhodamine B (RhB). A prism arrangement in a photon tunneling configuration was used for probing the dispersion, and two prisms (SF11, refractive index ~1.78) with 40 nm thick Ag films coated over their hypotenuse faces were brought close to each other to form the cavity, as illustrated in Figure 7a. Moreover, the position of the RhB in the cavity results in the formation of three antisymmetric plasmon (ASP) mode–exciton hybrid modes and splits the measured ASP transmission peak into three. Figure 7b shows the position of these peaks, where a permanent broad gap of 420 meV coupling is obtained, confirming the ultra-strong coupling between the exciton levels and the ASP mode.

Hakala T. et al. [120] demonstrated the emission of three energy branches in the strong coupling between the surface plasmon polaritons and J-aggregates Rhodamine 6G (R6G) sandwich structure, with a glass substrate, a 45 nm silver film in the middle, and a resist layer on top, with double vacuum Rabi splitting energies up to 230 and 110 meV for the 200 mM sample R6G concentrations. It is the first time that in the strongly coupled SPP-molecule system, each of the three energy branches can be converted into photons, and the finding shows great foreground in the field of multimode energy transfer and hybridization.

In all, the one-dye microcavity nanostructure system can be used to provide multimode hybrid interactions, which may inspire related exploring on multimode light–matter interactions and achieving some potential applications, such as multimode sensors and spectroscopy.

#### 3.1.3. Coupling Related to Two-Dye Microcavity Nanostructure

It is worth noting that current research leans towards more than two types of light components to form hybrid polaritons, one with strong dispersion and the other one nearly nondispersive, which can inspire related studies on hybrid light–matter interactions. To date, interactions between electronic states in organic states and plasmons have been undertaken intensively. In particular, when the emitter which is near the metal surface with the plasmon resonance, the spontaneous emission rate is modulated and the rate of energy transfer is changed [121,122]. However, most studies are conducted under the weak coupling region. On the other hand, pioneering studies have demonstrated that organic dye molecules with high oscillator strength form strong coupling region under plasmon excitation [123,124]. The coupling strength can be controlled through the plasmon energy and its line width, suggesting usability of a wide variety of organic molecules for achieving strong coupling.

Lidzey D. et al. [125] presented microcavities that can occur between the two cyanine dyes, with simultaneous strong coupling of the excitations of the individual dyes to a single cavity mode. The cyanine dye layers were spatially separated by a 100 nm-thick barrier layer of polystyrene to give an exciton separation of 140 nm. The cavity contained a dye concentration of approximately 2.3 × 10^20^ cm^−3^, where the 37 meV and 58 meV splitting branches were observed, whereas with a lower dye concentration of 1.2 × 10^20^ cm^−3^, the splitting branches were 18 meV and 44 meV. The exciton scattering rates were expected to be significantly enhanced because Frenkel excitons have large interaction cross-sections with molecular vibrations.

Melnikau D. et al. [126] introduced hybrid structures consisting of Au nanostars and J-aggregates of cyanine dyes, and Rabi splitting with an energy of up to 260 meV. In addition, the absorption spectrum of the complex hybrid system showed two pronounced dips at 590 and 642 nm (red curve), which corresponded to the maximum absorption wavelengths of the two different dyes (JC1 and S2165), and double Rabi splitting with the energies of 187 and 119 meV, as shown in Figure 8a. In addition, Figure 8b is the results from the model simulations [127] corroborated the experimental findings, when the position of the exciton resonance shifts to red or blue, which is relative to the maximum absorption of the nanostar, a unique asymmetric profile can be seen in the spectrum of the hybrid system.

Coles D. et al. [128] used strong coupling in an optical microcavity to mix the electronic transitions of two J-aggregated molecular dyes (TDBC and NK-2707) with a 200-nm-thick silver mirror, and used both non-resonant photoluminescence emission and photoluminescence excitation spectroscopy to show that hybrid polariton states act as an efficient and ultrafast energy-transfer pathway between the two exciton states. At normal incidence, the photon energy was 101 meV below the NK-2707 exciton (hereafter referred to ex1 for simplicity) and 250 meV below the TDBC exciton (hereafter ex2). A Rabi splitting energy of $h\Omega_{1}$ = 73 meV was determined between the LPB and MPB, whereas there was a larger splitting of $h\Omega_{2}$ = 155 meV between the MPB and UPB, consistent with the higher oscillator strength contributed from the TDBC J-aggregates.

It is worth noting that, apart from the interaction area length, the two-dyes microcavity nanostructure can increase the SPP-material interaction by creating a layer at the top of the interacting area, thereby decreasing the mode volume and preventing decay into the radiative mode, in which the upper exciton reservoir alone could be excited. This shows potential applications in integrated microcavity sensors and optical devices.

### 3.2. Multimode Coupling Related to Periodic Noble Metallic NPs- J-Aggregates Nanostructure

Organic semiconductor materials, such as J-aggregate [129,130,131] dyes, offer great potential because of their advantages in low-cost manufacturing and optical properties. J-aggregates support exciton states, which are electrically neutral electron/hole pairs generated by the absorption of photons. Excitonic states also exhibit strong nonlinear optical behavior that can be used to generate excitation sources for photon and transistor effects.

In addition, the periodic noble metallic nanoparticles (NPs) can be adjusted to modulate the plasmonic response of the nanostructure, therefore, the new periodic NPs monolayer nanostructure becomes a prime candidate for researchers exploring strong coupling phenomena.

Zhang K. et al. [132] demonstrated a nanostructured cavity with hybrid coupling among the molecular excitons, surface plasmon polaritons, and Fabry–Perot mode, where a J-aggregate-doped polyvinyl alcohol (PVA) layer was inserted between a silver grating and a thick silver film. They designed a nanostructure PVA with a thickness of 170 nm, inserted between a silver grating of 300 nm period and a 40 nm slit width, as can be seen in Figure 9a, the SPPs energy can be tuned through changing the period of the grating. In addition, for introduction of the excitons, the pure PVA was replaced with cyanine dye TDBC, as shown in Figure 9b. They observed three dispersive polariton bands with Rabi splitting energies of $h\Omega_{1}=110.6\;\mathrm{meV}$ and $h\Omega_{2}=122.5\;\mathrm{meV}$, as shown in Figure 9c, indicating the strong coupling between the SPPs mode and the PB1 or PB2 mode (there are two unique solutions, thus two hybrid modes emerging). Due to the non-dispersion of the Fabry–Perot modes and the exciton, these mixed modes hardly disperse with the plane wave vector, as shown in Figure 9c, the two modes PB1 and PB2 were labeled.

Wang H. et al. [133] constructed a hybrid system consisting of a 200 nm thick Au film patterned with a 2D square lattice gold nanohole array, and three different samples were prepared by spin-coating a uniform layer (300 nm thick) of sulforhodamine 101 (SR101) films with different concentrations, as can be seen in Figure 10a. Furthermore, by increasing the SR101 concentration, the Rabi splitting widened. The double Rabi splitting energies were 255 and 188 meV in steady-state transmission measurements, as illustrated in Figure 10b, and the dispersion was in excellent agreement with the typical signature of strong coupling. When the gold nanohole period is 380 nm, which matches the absorption of SR101 of the SPP resonance, the coupling strength maximizes, and when the plasmon mode periodicity is 350 nm, which matches the shoulder absorption of SR101, the double Rabi splitting is observable for the two periods.

Yang H. et al. [134] investigated plasmon−exciton−plasmon (PEP) couplings in Ag−J-aggregates−Ag (AJA) nanostructures using the finite element method, which can be seen in Figure 11a. The grating period is 500 nm, which is not excessively large so as to superpose the lattice resonance with the LSPR. The three new hybridized plexciton modes were obtained by adjusting the geometry and the incident angle. As shown in Figure 11b, the solid lines indicated three plexciton bands $E_{PEP}^{H},E_{PEP}^{M},$ and $E_{PEP}^{L}$ obtained based on Equation (16), where it was clearly observed that the solid lines matched well with the numerical results. The coupling of the SPP mode with $E_{J-LSPR}^{L}$ and $E_{J-LSPR}^{H}$ modes had splitting energies $h\Omega_{H}=55.3\;\mathrm{meV}$ and $h\Omega_{L}=52.5\;\mathrm{meV}$, respectively, which are smaller than the splitting energy between $E_{J-LSPR}^{L}$ and $E_{J-LSPR}^{L}$ (169 meV). They found that the resulting plexciton states are part-light and part-matter, where the light fraction is the sum of the SPP and LSPR modes.

Li M. et al. [135] demonstrated a classic oscillator coupled model consisting of a 300 nm thick SiO_2_ substrate and a 200 nm thick silver film with 100 nm diameter periodic hole arrays. The molecular layer doped with polyvinyl alcohol (PVA) to enhance the localized SPP electric field was deposited on the 40 nm thick top layer and in the holes, as illustrated in Figure 12a. Different periods of theoretical curves are plotted and was in good agreement with simulated results, which can be seen in Figure 12b. In the lower polaritons band, the slight inconsistency is due to the dominating SPP modes in the hybrid system. Furthermore, the detailed formation of a strong coupling was proposed based on three states: First, the interaction between the SPP (1,0) mode with molecular excitons generates the first Rabi splitting, and then with the adjustment of the system structure, the interaction between SPP (1,1) mode and the excitons generates the second anticrossing effect. It is the double Rabi splitting that draws attention to exploring the enhanced strong coupling effects. A maximum effective coupling strength of 0.316 and Rabi splitting value of 663 meV.

Evidently, both the selection of this type of J-aggregated molecular and grating period were dictated by the need to realize strong coupling, motivated by the exceptionally high oscillator strength and narrow resonances, even at room temperature. The corresponding work may inspire related studies on hybrid light–matter interactions and achieve potential applications in multimode lasers and optical micro-spectroscopy.

### 3.3. Multimode Coupling Related to Core–Shell Multicomponent Systems

Various interesting phenomena occur if the coupling between the cavity and the emitter is sufficiently large such that the characteristic interaction time exceeds all other decay channels. In this strong coupling regime, photon and matter excitations hybridize to form a new type of quasi particle. These polaritons are revealed by two new energy-shifted optical resonances. They possess both light- and matter-like properties, which make them suitable for a wide range of applications. Here, the controlled strong coupling of a single emitter to a cavity allows quantum-state mapping of a localized emitter qubit to a moving photon qubit. For this type of application, it is essential to achieve strong coupling of a single emitter.

Melnikau D. et al. [92] demonstrated a single hybrid structure with multiple spectral features, which was integrated with three different components, including core–shell Au@Ag nanorods (NRs), and two different dyes (5,5′,6,6′-tetra-chloro-1,1′,3,3′-tetraethyl-imidacarbocyanine iodide(TCI) and 2-[3-[1,1-dimethyl-3-(4-sulfobutyl)-1,3-dihydro-benzo[e]-indol-2-ylidene]-propenyl]-1,1-dimethyl-3-(4-sulfobutyl)-1H-benzo[e]indolium hydroxide, (DBI)), as shown in Figure 13a, leading to a strong collective exciton-plasmon coupling. Figure 13b displays the theoretical results (solid lines) for the Au@Ag NRs with the two dyes fitted to the experimental data obtained for the hybrid plexciton states in the case of double Rabi splitting, and the estimated Rabi splitting values are 175 and 163 meV. All three components were involved in the strong interaction, and the total value of 338 meV reflects the onset of collective extended energy splitting between the states of the lower and upper plexcitons.

Despite the fact that microcavities are considered to be an excellent alternative to plasmonic nanostructures for achieving strong coupling, the core–shell multicomponent systems show magneto-optical activity for hybridized plexciton states resulting from strong exciton–plasmon coupling in nanostructures, in which the plasmonic component induces magnetic properties in nonmagnetic organic fluorophores. This is significant for the development of new sensing systems based on magneto-optical activity, while expanding the portfolio of materials that can be used for optical information storage and processing.

Different multimode coupling with their applications is shown in Table 1.

## 4. Conclusions and Development Trends

In general, surface plasmons have become a popular topic in the field of photonics owing to their distinctive optical properties [136,137]. This is particularly evident on the upper surface in studying the coupling between surface plasmons and excitons. This paper reviewed the research progress in the field based on multimode coupling related to TMDs microcavity nanostructure, single-dye microcavity nanostructure, two-dyes microcavity nanostructure and core–shell multicomponent systems. This paper is introduced from a theoretical point of view. It has been shown that multimode coupling can not only reduce the size of quantum devices, but also has preliminarily proven the superiority to regulating the interaction between light and matter.

However, there remain numerous unknown problems regarding multimode coupling based on surface plasma nanostructures, which should be explored further in terms of the following aspects:

(1) Strong coupling (the signature is a single Rabi splitting) can induce photon electricity and photon thermal processes, whereas multimode coupling is more complicated than general coupling, and may lead to a new series of energy conversion processes, such as photon–thermal–electricity and photon–electricity–thermal processes induced by surface plasmons. In addition, it provides a new route for further exploration of the physical mechanism of surface plasmons, which bears potential scientific applications for the development of new nano-optical devices.

(2) Strong coupling can be demonstrated including ultrafast tuning of strongly coupled metal–molecular aggregates via femtosecond pumping [138], dynamical modification of the polariton composition [139] and UV illumination of silver nanoparticle arrays [140]. However, there are few studies in this field of new physics, which concern potential phase changes, such as structure-induced phase changes and temperature-induced phase changes. Moreover, it may provide an exciting topic for future research.

(3) With the help of a database, we discovered that reports based on multimode splitting are limited, and that single Rabi splitting is the mainstream in current research fields. If the conditions are appropriately changed, future research hotspots would focus on whether a higher-mode process could occur.

(4) In practical applications, research into quantum information processing, higher-order processes, and nonlinear optics between surface plasmons and matter, are expected to be further discussed in terms of the physical essence of interactions.

(5) As an emerging discipline, machine learning applications in plasmonics have attracted considerable attention [141,142,143], such as both the forward prediction of far-field optical properties and the inverse prediction of on-demand dimensional parameters of nanoparticles [144], and the prediction of color of nano-structured surfaces through either the nanoparticle geometric parameters, or laser parameters [145]. However, there are few examples by using machine learning to study the multimode strong coupling, which may be one of the future focuses.

The multimode coupling of plasmons and excitons based on metal nanostructures have greatly promoted the development of strong coupling. In particular, it contains considerable potential for application in the design and improvement of organic photoelectric devices such as organic luminescence and organic batteries [146,147]. Therefore, surface plasma optics, produced by the combination of quantum optics and surface plasmon optics, may serve as a new development direction. This will not only increase the depth of research regarding the basic properties of surface plasmon excitons but also provide directions for solving current problems in quantum optics research.

## Figures and Tables

**Figure 1 nanomaterials-12-01242-f001:**
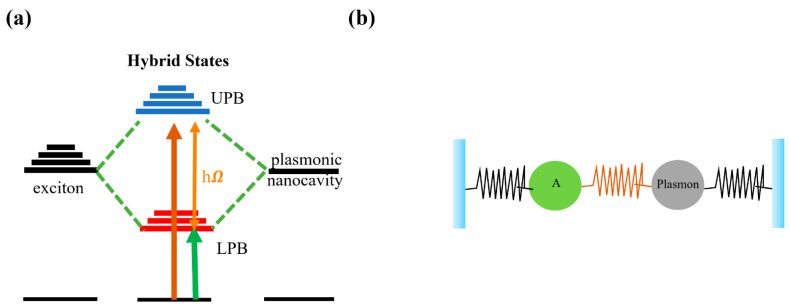
(**a**) Schematic of the energy states of the double coupling of the plasmon mode with the exciton. (**b**) Strong coupling regime illustrated by two coupled oscillators. Adapted with permission from ref. [74]. Copyright 2010 AIP Publishing.

**Figure 2 nanomaterials-12-01242-f002:**
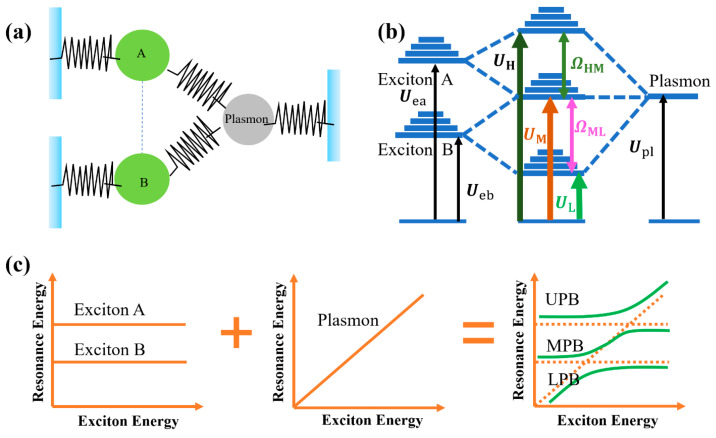
(**a**) Schematic diagram of the three-coupled oscillator model. (**b**) Schematic diagram of plasmon and exciton multimode coupling energy level splitting. Adapted with permission from ref. [92]. Copyright 2019 American Chemical Society. (**c**) Double anticrossing dispersion curves of plasmon–exciton multimode coupling.

**Figure 3 nanomaterials-12-01242-f003:**
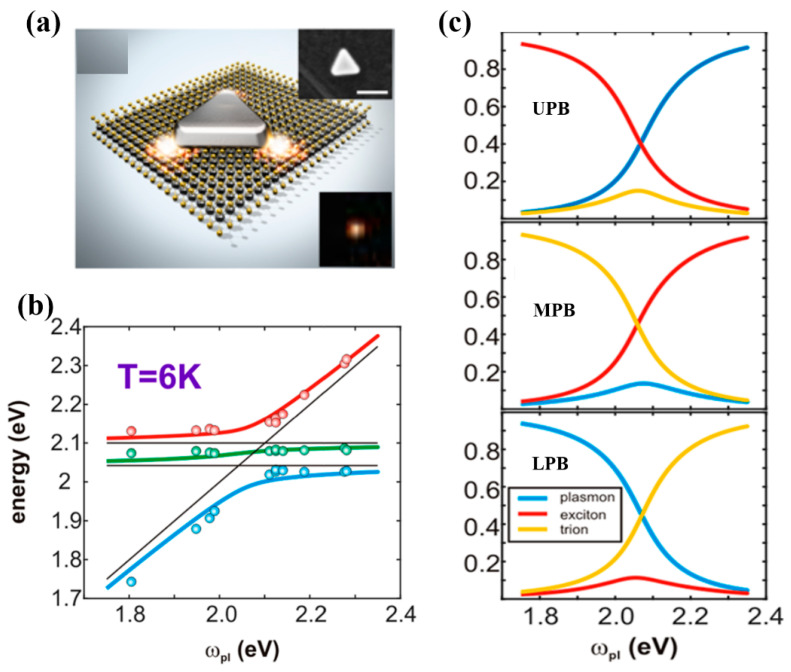
(**a**) Schematic of the silver nanoprism–WS2 hybrid system. (**b**) Eigen energies as a function of plasmon resonance position extracted from dark-field spectra of hybrid systems of various sizes and plasmon−exciton−trion detuning at T = 6 K. (**c**) Hopfield coefficients for plasmon, exciton, and trion contributions to UPB, MPB, and LPB states as a function of plasmon resonance. Adapted with permission from ref. [114]. Copyright 2018 American Chemical Society.

**Figure 4 nanomaterials-12-01242-f004:**
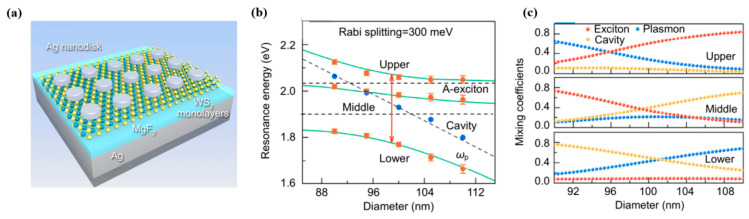
(**a**) Schematic of the Ag–WS_2_ heterostructure, where Ag nanodisks were fabricated on WS_2_ monolayers by EBL. (**b**) Energies of reflectivity dips as a function of the nanodisk diameter extracted from the reflectivity spectrum. (**c**) Hopfield coefficients for plasmon, exciton, and microcavity contributions to upper, middle, and lower hybrid states as a function of diameter, calculated using the three-coupled oscillator model. Adapted with permission from ref. [116]. Copyright 2019 Institute of Optics and Electronics, Chinese Academy of Sciences.

**Figure 5 nanomaterials-12-01242-f005:**
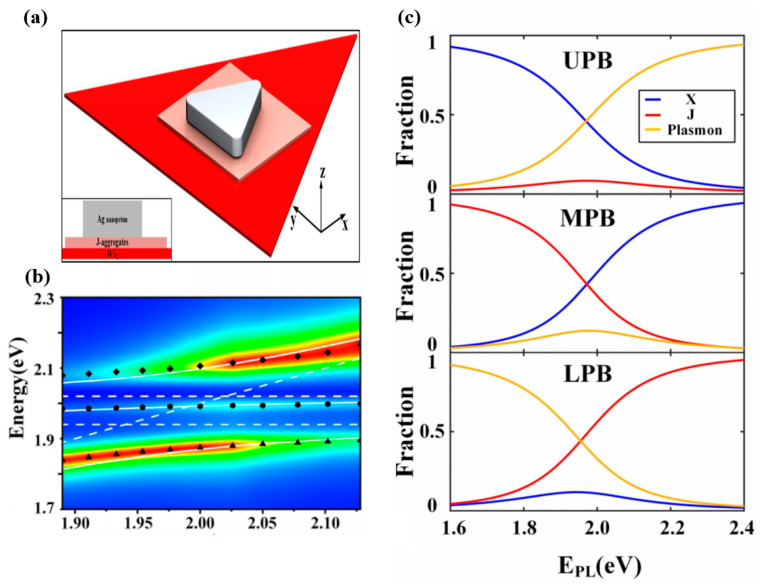
(**a**) Schematic of the Ag–J-aggregates–WS_2_ nanostructure. The inset shows the cross section of the hybrid nanostructure with a Ag nanoprism thickness of 10 nm. (**b**) Scattering cross sections spectra of J-aggregates and WS_2_ coupled with Ag nanoprisms. The black scattered-diamonds, scattered-circles and scattered-triangles represent simulated UPB, MPB and LPB as a function of plasmon resonance position. (**c**) Weighting efficiencies for LSPR mode, the exciton of WS_2_ and J-aggregates exciton contributions to UPB, MPB and LPB states as a function of plasmon resonance. Adapted from ref. [117].

**Figure 6 nanomaterials-12-01242-f006:**
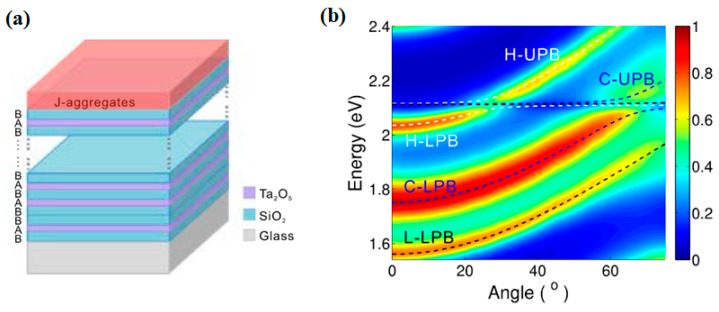
(**a**) Schematic of the photonic quasicrystal covered by the organic semiconductor. (**b**) Two-dimensional dispersion map of the sample covered by the J-aggregates. The dashed lines are the calculated results fitting the polariton bands. Adapted with permission from ref. [118]. Copyright 2016 Optical Society of America.

**Figure 7 nanomaterials-12-01242-f007:**
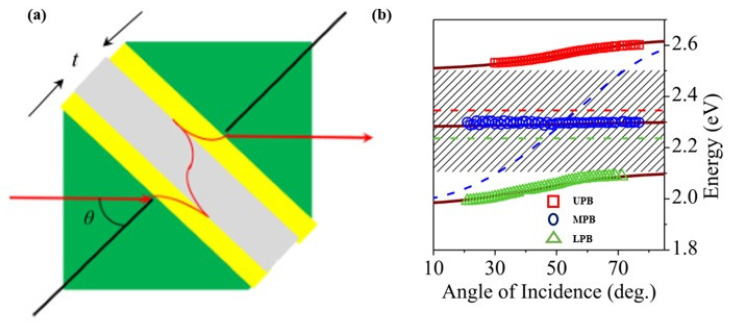
(**a**) Schematic of the photon tunneling based configuration. (**b**) Mapped peak position showing the dispersion of the ASP–exciton hybrid modes. Adapted with permission from ref. [119]. Copyright 2017 American Institute of Physics.

**Figure 8 nanomaterials-12-01242-f008:**
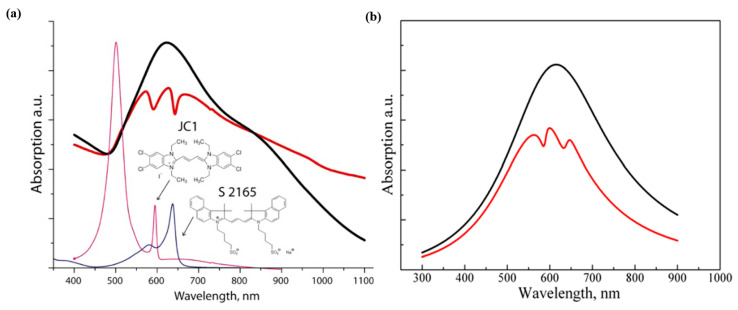
(**a**) Absorption spectra of gold nanostars (balck curve), pristine J-aggregates of JC1 (magenta curve) and S2165 (blue curve), and the hybrid structure (red curve). (**b**) Theoretical extinction spectra of gold nanostars (black curve) and their hybrid structure with J-aggregates (red curve). Adapted with permission from ref. [126]. Copyright 2013 IEEE.

**Figure 9 nanomaterials-12-01242-f009:**
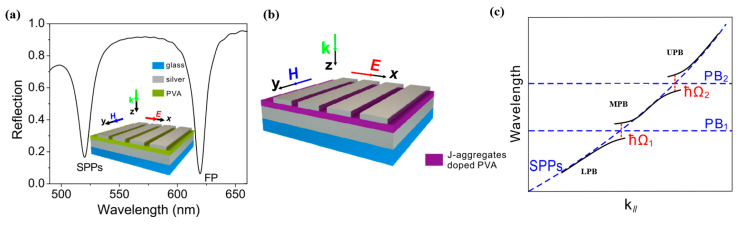
(**a**) The calculated reflection spectrum of the nanostructure with PVA inserted between a silver grating and a thick silver film. The inset gives a schematic description of the nanostructure. (**b**) A schematic description of the TDBC J-aggregates doped nanostructure. (**c**) Sketch of the interactions among SPPs mode, PB1 mode, and PB2 mode. Adapted with permission from ref. [132]. Copyright 2016 American Institute of Physics.

**Figure 10 nanomaterials-12-01242-f010:**
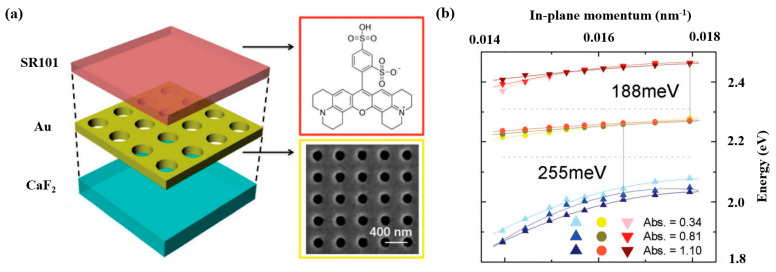
(**a**) Schematic of the hybrid exciton–plasmon system. The red upper part illustrates the chemical structure of SR 101 dyes, while an SEM image of the Au nanohole array is shown in the yellow box. (**b**) The energy dispersion curves, where the three bands were measured by changing the SR101 concentration. Adapted with permission from ref. [133]. Copyright 2017 WILEY-VCH Verlag GmbH & Co. KGaA, Weinheim.

**Figure 11 nanomaterials-12-01242-f011:**
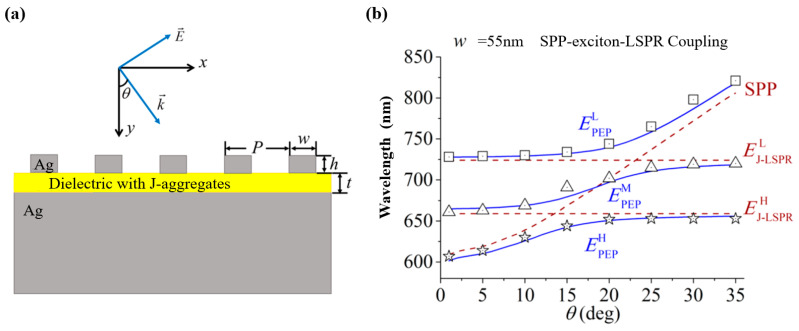
(**a**) Schematic diagram of the Ag–J-aggregates–Ag nanostructure. (**b**) Dispersive curves of three plexcimon modes in the Ag−J-aggregates−Ag nanostructure. Adapted with permission from ref. [134]. Copyright 2017 American Chemical Society.

**Figure 12 nanomaterials-12-01242-f012:**
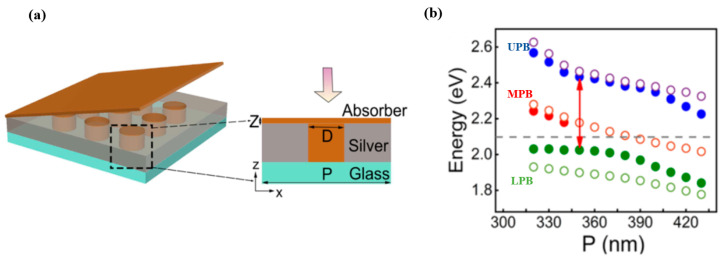
(**a**) Schematic diagram of the multiple plasmonic modes and a single molecular exciton hybrid system. (**b**) Dispersive curves of the hybrid polaritonic energies based on the coupled model in the period 320–430 nm. Adapted with permission from ref. [135]. Copyright 2020 American Chemical Society.

**Figure 13 nanomaterials-12-01242-f013:**
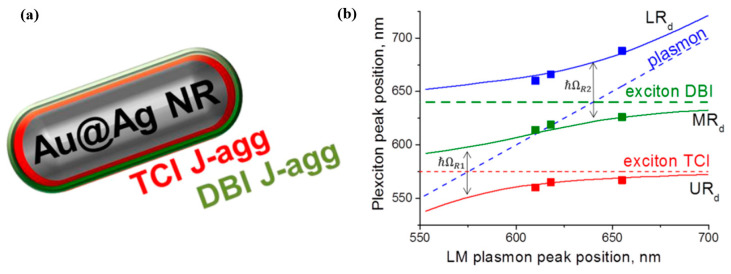
(**a**) Schematic diagram of the single hybrid structure, which is integrated by three different components, such as core–shell Au@Ag nanorods (NRs) and two different dyes. (**b**) Positions of LRd (blue squares), MRd (green squares), and URd (red squares) in the experimental extinction spectra of the hybrid structure of Au@Ag NRs and J-aggregates of TCI and DBI dyes with double Rabi splitting, as a function of the plasmon peak position in the spectra of bare NRs. Adapted with permission from ref. [92]. Copyright 2019 American Chemical Society.

**Table 1 nanomaterials-12-01242-t001:** Different multimode coupling with their applications.

Different Multimode Coupling		Applications
Multimode Coupling related to microcavity nanostructure	Coupling related to TMDs microcavity nanostructure	Optical modulators at the nanoscale and polaritonic devices based on ultrathin materials, etc.
	Coupling related to single-dye microcavity nanostructure	Multimode sensors and spectroscopy, etc.
	Coupling related to two-dyes microcavity nanostructure	Integrated microcavity sensors and optical device, etc.
	Coupling related to periodic noble metallic nanoparticles- J-aggregates nanostructure	Multimode lasers and optical micro-spectroscopy, etc.
Multimode coupling related to core–shell multicomponent systems		Nanoscale optical information storage and processing, etc.

## Data Availability

The study did not report any data.

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
