# Peer review of "Principle and Applications of Multimode Strong Coupling Based on Surface Plasmons"

_nanomaterials, 2022, doi:10.3390/nano12081242_

Round 1

Reviewer 1 Report

Please revise the Eq.2.

Please cite the recently published papers 2019 onward (at least 30%)

Author Response

1. Please revise the Eq.2.

Reply: Thanks for your suggestion. We have revised the Eq.2 in our revised manuscript, which is shown in red on Page 3 in line 129.

2. Please cite the recently published papers 2019 onward (at least 30%)

Reply: Thanks for your suggestion. We have cited the recently published paper 2019 onward in our revised manuscript, which is more than 30%.

Reviewer 2 Report

The presented review begins in an unusual way, namely with a paragraph describing the general requirements for introductions to articles. Most likely, this was the result of an oversight by the authors (as well as the formula symbols that are not aligned with the text of the article). Nevertheless, I will also allow myself to formulate the general requirements that, in my opinion, a review should meet. A review is a manuscript in which a certain research area should be surveyed in a competent manner, based on various sources. Moreover, the survey should be presented in such a way that the main reviewed results and approaches are understandable, without reading the original articles, not only to a narrow specialist in the problem, but also to physicists working in related fields.

Unfortunately, in my opinion, this review does not meet these requirements. Consider, for example, section 2, which presents a description of the theoretical approach. Contrary to expectation, it begins not with the formulation of the problem of theoretical studies (i.e., what parameters need to be calculated and what phenomena should be explained), but with the fact that a formula is given for some Dicke Hamiltonian (1), which, according to the authors, “displays the interaction of a single mode radiation field with N-identical two-level systems under the dipole approximation…”. What is the Dicke model and how well does the Hamiltonian proposed by him describe the interaction of light and matter? With what kind of mater? It is not clear what particular problems within the general study can be addressed with this Hamiltonian? Moreover, the authors make no reference to this Hamiltonian or to its consequences in the subsequent text of the article.

 The Hamiltonian is an operator that specifies the evolution of the quantum states of a particle or a set of particles and the spectrum of possible energy values ​​of the system. What particles are we describing with expression (1)? What equation does the Dicke Hamiltonian form (what is the potential energy of the system) and what is its solution? If there is no solution, then what properties of the solution or values ​​of the energy of the system can be extracted from the form of the given Hamiltonian? The authors do not answer these questions, suggesting that the reader should guess for himself. But this is quite hard to do, since the description of most of the symbols in equation (1) is missing. What do the symbols a+, a, J+, J-, Jz mean?

 Are symbols a, a+ (apparently creation-annihilation operators) from equation (1) the same as those used in equation (2) (it has “a” and “a with the Hermitian conjugation symbol”) or not?

Regarding the Jaynes Cummings Hamiltonian defined by expression (2), the same questions can be asked. What is the Jaynes Cummings model? What equation does this Hamiltonian form (what is the potential energy of the system) and what is the solution to this equation?

Based on the form of the Hamiltonian (2), it can be assumed that its first term specifies the SPP energy, the second term defines the energy of the atom, and the third term describes the interaction of the atom with the plasmon. Is it so? Or not? If this is so, then why is it about the interaction of SPP with an atom, and not with an exciton? Maybe expression (2) still refers to an exciton? Indeed, as the authors state “Diagonalization of the Hamiltonian in Eq. (2), the classical coupled harmonic oscillator model is often used to predict the upper polariton branches (UPB) and lower polariton branches (LPB)”. As far as it is clear from what follows, UPB and LPB are mixed (hybridized) states between SPP and the exciton radiation field. Does this mean that the original expression (2) also refers to excitons? Why, then, is the Pauli matrix formalism used to describe the corresponding term of the Hamiltonian (the second term in (2))? This formalism is suitable only for describing fermions, while an exciton is a boson.

Further, in expression (7), one more Hamiltonian is written without discussing the corresponding equation and its solution. What for?

Next, the authors try to consider the problem of the interaction of an exciton with a plasmon from the classical point of view, using the model of coupled oscillators. In most cases, this approach is very productive, since coupled mechanical pendulums are very illustrative, the corresponding differential equations are simple, and the properties of their solutions are intuitive. Dependences obtained on the basis of those solutions give a good qualitative (and sometimes quantitative) description of the spectra of various atomic systems or metastructures. In this regard, I would like to see in the review a discussion of the question of how the two models (quantum and classical) complement each other in describing the interaction of an exciton with a plasmon and what are their limitations. But this discussion is not included in the review.

It is not very clear why the article does not provide a system of oscillation equations for two coupled oscillators (it is given in the next paragraph, but for the case of three oscillators). In this regard, the phrases in the text “… the coupled oscillator equation can be solved easily…”, “results of the coupled-oscillator model are…” look strange. Indeed, the equation of oscillations of two coupled pendulums is easily solved and can be written analytically in explicit form. In particular, on the basis of this solution, the further arguments of the authors about the frequencies omega+, omega-, strong and weak coupling of modes, as well as comparison of the classical and quantum results would be more understandable to the reader. But this solution is not included in the review.

Concerning the system of equations of three coupled oscillators in the next section of the review, one can notice the following. These equations include the quantities x_pl, x_a, x_b, which the authors of the review say: “where x_pl, x_a, x_b plasmon, exciton A oscillator, and exciton B oscillator, respectively”. In my opinion, such an explanation represents an outrageous disrespect for the reader. What parameter of the plasmon, excitons A and exciton B do you mean? The amplitude of the electromagnetic field? Amplitude of Psi-functions? Level population? Something else?

In general, it seems that, despite the indices “pl”, “a”, “b”, the system of equations (13-15) most probably describes a mechanical system. This, in particular, is indicated by the fact that the authors call the quantities F_pl, F_a, F_b included in the right side of the equations “driving forces due to the external source”. Are those really mechanical forces measured in Newtons? What units should x_pl then have? Judging by the units in the following formula for the scattering cross section, the product x_pl*F_pl must have units of m^3. Therefore, x_pl should be a “mysterious” quantity with the units of m*s^2/kg?

The authors then again turn to quantum mechanics and write down the Hamiltonian (17). Judging by the matrix notation, it corresponds to Heisenberg's approach. However, in order to write down such a Hamiltonian, one must actually know explicitly the Psi-functions of the system. What are they? Before writing expression (17), nothing is said about them. Maybe it's "unperturbed basis functions phi_pl, phi_a, phi_b" that are mentioned later when discussing equations (18-20)? Or not?

The review then talks about Psi-functions for the hybrid UPB, MPB and LPB modes. However, judging by the equations (18-20) given for them, they are simply linear combinations of the basis functions phi_pl, phi_a, phi_b. How is this possible for those functions that should appear as a result of hybridization of the basic ones?

Finally, in the last paragraph of the theoretical section "Scattering Spectra" in expressions (21-24), the value mu_pl appears again, which was previously mentioned only once in the expression with the Hamiltonian (7), where it was indicated that "mu_pl is the plasmon dipole operator". How can the operator be equal to a simple algebraic expression (for example, (24)), which defines the scattering spectrum? Or is it something else? The eigenvalue of the mu_pl operator? In what equation?

What does mu_em mean in (23), which is not mentioned anywhere else in the paper?

Doesn't expression (16) given earlier in paragraph 2.2 for the scattering cross section define the scattering spectrum? Why use (24) then?

Where are the examples of calculating the scattering spectra of any model materials with a description of the features of these spectra, which could be further compared with the experimental data to make a conclusion about the suitability of the above theoretical description?

Perhaps the answers to the above questions seem obvious to a specialist who is deeply familiar with the content of all the articles mentioned by the authors of the review. However, why does such an advanced specialist need this review? And for specialists in related fields, most of the theoretical descriptions of this review are completely incomprehensible and, therefore, useless.

Further in section 3 the results of experimental studies are described. There are much fewer obvious "inconsistencies" here. However, they are. For example, the question of why the experimental data obtained by different authors can be interpreted as the result of the interaction of excitons and plasmons is not discussed anywhere in this section.

The caption to figure 3 says “Energies of reflectivity dips as a function of the nanodisk diameter extracted from the reflectivity spectrum”. What do the original reflectance spectra look like and how is the data extraction procedure performed?

In Fig. 8 (unfortunately, this is perhaps the only example of a spectrum), only two of the four curves are identified. Are the presented curves experimental or still calculated? If experimental, where is the comparison with the calculated curves?

In general, section 3 is perceived as some compilation of information from the published works of other researchers, without reasonable assessments, generalizations and logical conclusions.

Taking account of the above, I consider the publication of the review to be inappropriate.

Author Response

Reviewer2

  1. Consider, for example, section 2, which presents a description of the theoretical approach. Contrary to expectation, it begins not with the formulation of the problem of theoretical studies (i.e., what parameters need to be calculated and what phenomena should be explained), but with the fact that a formula is given for some Dicke Hamiltonian (1), which, according to the authors, “displays the interaction of a single mode radiation field with N-identical two-level systems under the dipole approximation…”. What is the Dicke model and how well does the Hamiltonian proposed by him describe the interaction of light and matter? With what kind of mater? It is not clear what particular problems within the general study can be addressed with this Hamiltonian? Moreover, the authors make no reference to this Hamiltonian or to its consequences in the subsequent text of the article.

Reply: Thanks for your comments. Section 2 is “basic principle of multimode strong coupling”, which is help reader for a better understanding about the strong coupling, especially multimode strong coupling. Dicke Hamiltonian is “a model that is essential and crucial in quantum optics, which is in the description of the systems relating to the interaction of light and materials, and as the simplest model”. In addition, we have added the related reference No.54-56 in our revised version, which is shown in red on Page 3 in line 107. As the Dicke model is a fundamental model of quantum optics, which describes the interaction between light and matter. In the Dicke model, the light component is described as a single quantum mode, while the matter is described as a set of two-level systems. When the coupling between the light and matter crosses a critical value, the Dicke model shows a mean-field phase transition to a superradiant phase. This transition belongs to the lsing universality class and was realized experimentally in cavity quantum electrodynamics experiments [Plos one, 2020, 15(9): e0235197]. And the matter are the common materials, such as noble nanoparticles, J-aggregates or two-dimensional transition metal dichalcogenides, etc. And the Hamiltonian can be performed for any type of coupling [The European Physical Journal B, 1998, 6: 201], coupled to one (or multiple) baths of free fermions [Physical Review B, 2018, 97:205405], or obtained the standard Dicke Hamiltonians in the dipole and in the Coulomb gauges which both bilinear in the bosonic operators [Physical Review A, 2020, 102:023718]. And we have added the related references No. 58-62 to this Hamiltonian or to its consequences in the subsequent text of the article in our revised manuscript, which is shown in red on Page 3 in line 122.

  1. The Hamiltonian is an operator that specifies the evolution of the quantum states of a particle or a set of particles and the spectrum of possible energy values ​​of the system. What particles are we describing with expression (1)? What equation does the Dicke Hamiltonian form (what is the potential energy of the system) and what is its solution? If there is no solution, then what properties of the solution or values ​​of the energy of the system can be extracted from the form of the given Hamiltonian? The authors do not answer these questions, suggesting that the reader should guess for himself. But this is quite hard to do, since the description of most of the symbols in equation (1) is missing. What do the symbols a+, a, J+, J-, Jz mean?

Reply: Thanks for your comments. The N identical two-level atoms immersed in a one-mode Dicke Hamiltonian was described by expression (1) [Physical Review A 2011, 83 (5), 051601]. And we have added the related representation about Dicke Hamiltonian, such as the Hamiltonian form and its solution, so as the symbols a+, a, J+, J-, Jz mean in our revised version, which is shown in red on Page 3 in line 107-122.

  1. Are symbols a, a+ (apparently creation-annihilation operators) from equation (1) the same as those used in equation (2) (it has “a” and “a with the Hermitian conjugation symbol”) or not? Regarding the Jaynes Cummings Hamiltonian defined by expression (2), the same questions can be asked. What is the Jaynes Cummings model? What equation does this Hamiltonian form (what is the potential energy of the system) and what is the solution to this equation?

Reply: Thanks for your comments. Symbols a, a+, actually (apparently creation-annihilation operators) from equation (1) are the same as those used in equation (2) (it has “a” and “a with the Hermitian conjugation symbol”). The Jaynes-Cummings (JC) model “is the simplest but nontrivial model for depicting interaction of a two-level atom with an electromagnetic field at resonance” [Journal of Mathematical Physics, 2009,50,103523]. And the equation was considered a two-state system coupled to a single mode of the radiation field with total Hamiltonian , where  and  are the raising and lowering operators of the two-state system, and are the creation and annihilation operators of the field mode, and the coupling term is in the Jaynes-Cummings form. This model describes, e.g., the interaction between a two-level atom and a mode of the radiation field in the electric dipole and rotating wave approximation, and the detailed explanation about Jaynes Cummings model can be found in reference No. 64 [Journal of Modern Optics, 1993, 40(7): 1195] and No. 65 [Physical Review A, 2010, 82(6): 062114] in our revised version, which is shown in red on Page 3 in line 123-125.

  1. Based on the form of the Hamiltonian (2), it can be assumed that its first term specifies the SPP energy, the second term defines the energy of the atom, and the third term describes the interaction of the atom with the plasmon. Is it so? Or not? If this is so, then why is it about the interaction of SPP with an atom, and not with an exciton? Maybe expression (2) still refers to an exciton? Indeed, as the authors state “Diagonalization of the Hamiltonian in Eq. (2), the classical coupled harmonic oscillator model is often used to predict the upper polariton branches (UPB) and lower polariton branches (LPB)”. As far as it is clear from what follows, UPB and LPB are mixed (hybridized) states between SPP and the exciton radiation field. Does this mean that the original expression (2) also refers to excitons? Why, then, is the Pauli matrix formalism used to describe the corresponding term of the Hamiltonian (the second term in (2))? This formalism is suitable only for describing fermions, while an exciton is a boson.

Reply: Thanks for your comments. The Hamiltonian (2) describes the interaction between a two-level atom and a mode of the radiation field in the electric dipole and rotating wave approximation [Physical Review A, 2010, 82(6): 062114], in fact, it can be assumed that its first term specifies the SPP energy, the second term defines the energy of the exciton, and the third term describes the interaction of the exciton with the plasmon. And expression (2) also refers to an exciton or atom. In addition, it also shows the original expression (2) referring to excitons. In a mixed coupled system, it is composed of SPP, excitons or emitters. And the Pauli matrix formalism used to describe the corresponding term of the Hamiltonian can be obtained from reference [Phys. Status. Solidi B, 2018,1800185] and [Nanoscale, 2019, 11:14540].

  1. Further, in expression (7), one more Hamiltonian is written without discussing the corresponding equation and its solution. What for?

Reply: Thanks for your comments. The reason why one more Hamiltonian in expression (7) is used for introducing the external electric field , which acts on the plasmon, while the direct driving of the emitter by the external field can be ignored, and the corresponding equation and its solution are not the focus in this section.

  1. Next, the authors try to consider the problem of the interaction of an exciton with a plasmon from the classical point of view, using the model of coupled oscillators. In most cases, this approach is very productive, since coupled mechanical pendulums are very illustrative, the corresponding differential equations are simple, and the properties of their solutions are intuitive. Dependences obtained on the basis of those solutions give a good qualitative (and sometimes quantitative) description of the spectra of various atomic systems or metastructures. In this regard, I would like to see in the review a discussion of the question of how the two models (quantum and classical) complement each other in describing the interaction of an exciton with a plasmon and what are their limitations. But this discussion is not included in the review.

Reply: Thanks for your suggestion. We have added the describing about the interaction of an exciton with a plasmon and what are their limitations in our revised version, which is shown in red on Page 4 in line 157-165.

  1. It is not very clear why the article does not provide a system of oscillation equations for two coupled oscillators (it is given in the next paragraph, but for the case of three oscillators). In this regard, the phrases in the text “… the coupled oscillator equation can be solved easily…”, “results of the coupled-oscillator model are…” look strange. Indeed, the equation of oscillations of two coupled pendulums is easily solved and can be written analytically in explicit form. In particular, on the basis of this solution, the further arguments of the authors about the frequencies omega+, omega-, strong and weak coupling of modes, as well as comparison of the classical and quantum results would be more understandable to the reader. But this solution is not included in the review.

Reply: Thanks for your comments. We have updated Figure 1(b), which is strong coupling regime by two coupled oscillators, and the equation of oscillations of two coupled pendulums is easily solved and can be written analytically in explicit form. And we have improved the explanation of the frequencies omega+, omega-, strong and weak coupling of modes, as well as comparison of the classical and quantum results in our revised version, which is shown in red on Page 5 in line 190-219.

  1. Concerning the system of equations of three coupled oscillators in the next section of the review, one can notice the following. These equations include the quantities x_pl, x_a, x_b, which the authors of the review say: “where x_pl, x_a, x_b plasmon, exciton A oscillator, and exciton B oscillator, respectively”. In my opinion, such an explanation represents an outrageous disrespect for the reader. What parameter of the plasmon, excitons A and exciton B do you mean? The amplitude of the electromagnetic field? Amplitude of Psi-functions? Level population? Something else?

Reply: Thanks for your comments. These equations include the quantities x_pl, x_a, x_b, and the description of “where x_pl, x_a, x_b plasmon, exciton A oscillator, and exciton B oscillator, respectively”, we exampled this representation from Ref.[Optics Express, 2019,27(12):16613]. As we can obtain in section 3, different microcavities which composed of different materials, such as two-dimensional transition metal dichalcogenides (TMDs) (as exciton), different dyes (as exciton or plasmon), J-aggregates (as exciton) and noble metal nanoparticles (as plasmon), so we cannot confirm the specific parameter of the plasmon, excitons A and exciton B for the varies microsystem.

  1. In general, it seems that, despite the indices “pl”, “a”, “b”, the system of equations (13-15) most probably describes a mechanical system. This, in particular, is indicated by the fact that the authors call the quantities F_pl, F_a, F_b included in the right side of the equations “driving forces due to the external source”. Are those really mechanical forces measured in Newtons? What units should x_pl then have? Judging by the units in the following formula for the scattering cross section, the product x_pl*F_pl must have units of m^3. Therefore, x_pl should be a “mysterious” quantity with the units of m*s^2/kg?

Reply: Thanks for your comments. The system of equations (13-15, updates No.16-18 in our version) describe a mechanical system, and we have quoted these equations from different research groups, and we consider “those really mechanical forces measured in Newtons”. And “units should x_pl then have”, “x_pl quantity with the units of m*s^2/kg” are all not the focus in our manuscript, and we suggest the related references [Optics Express, 2010, 18(23):23633] and [Optics Express, 2019, 27(10):16614] to the reviewer for further understanding.

  1. The authors then again turn to quantum mechanics and write down the Hamiltonian (17). Judging by the matrix notation, it corresponds to Heisenberg's approach. However, in order to write down such a Hamiltonian, one must actually know explicitly the Psi-functions of the system. What are they? Before writing expression (17), nothing is said about them. Maybe it's "unperturbed basis functions phi_pl, phi_a, phi_b" that are mentioned later when discussing equations (18-20)? Or not?

Reply: Thanks for your comments. The interaction Hamiltonian for a three-coupled oscillator model can be found in references [Nano. Lett.,2018, 18:1777], [Physical Review B, 1998, 58(23):15367], and [Opto-Electronic Advances, 2019, 2(5): 190008], and we have cited these references in our revised version, which is shown in red on Page 8 in line 314. And the Psi-functions of the system is actually mentioned later when discussing equations (18-20, updates No.21-23) in our manuscript.

  1. The review then talks about Psi-functions for the hybrid UPB, MPB and LPB modes. However, judging by the equations (18-20) given for them, they are simply linear combinations of the basis functions phi_pl, phi_a, phi_b. How is this possible for those functions that should appear as a result of hybridization of the basic ones?

Reply: Thanks for your comments. The equations (18-20, updates No.21-23 in our version) are not simply linear combinations of the basis functions phi_pl, phi_a, phi_b, and those functions appear as a result of hybridization can be obtained in our revised version, which is shown in red on Page 8-9 in line 318-334.

  1. Finally, in the last paragraph of the theoretical section "Scattering Spectra" in expressions (21-24), the value mu_pl appears again, which was previously mentioned only once in the expression with the Hamiltonian (7), where it was indicated that "mu_pl is the plasmon dipole operator". How can the operator be equal to a simple algebraic expression (for example, (24)), which defines the scattering spectrum? Or is it something else? The eigenvalue of the mu_pl operator? In what equation?

Reply: Thanks for your comments. In previous publication [Optics Express, 1010,18:23633], a version of the coupled-oscillator equations with different coupling terms in our version with Eq. (26), and the coupling terms used in Eqs. (8) and (10) are preferred, which is due to the equation for the leading choice of the coupling term is derived from a quantum mechanical model in the linear and classical limitation, and if one uses the coupling terms in Eq. (24, updates No.27 in our version), the steady-state solution is different only in the last term of the denominator, we have added the related presentation in our revised version, which is shown in red on Page 9 in line 355-356.

  1. What does mu_em mean in (23), which is not mentioned anywhere else in the paper?

Reply: Thanks for your comments. We have defined the mu_em (23, updates No.26 in our version) in our manuscript on Page 3 in line 135 while we used the wrong format of formula, and we have corrected in our revised version, which is shown in red on Page 3 in line 135-136.

  1. Doesn't expression (16) given earlier in paragraph 2.2 for the scattering cross section define the scattering spectrum? Why use (24) then?

Reply: Thanks for your comments. Expression (16, updates No.19 in our version) given in paragraph 2.2 is the scattering cross-section, while Expression (24, updates No.27 in our version) represents the plasmon dipole.

  1. Where are the examples of calculating the scattering spectra of any model materials with a description of the features of these spectra, which could be further compared with the experimental data to make a conclusion about the suitability of the above theoretical description?

Reply: Thanks for your comments. The examples of calculating the scattering spectra of any model materials with a description of the features of these spectra can be obtained from [Optics Express, 2010, 18(23): 23633]、[American Journal of Physics,2002,70(1)] and [Nature Mater, 2009,8:758], which could be further compared with the experimental data to make a conclusion about the suitability of the above theoretical description, and we have cited these references in our revised manuscript, which is shown in red on Page 9 in line 361.

  1. Perhaps the answers to the above questions seem obvious to a specialist who is deeply familiar with the content of all the articles mentioned by the authors of the review. However, why does such an advanced specialist need this review? And for specialists in related fields, most of the theoretical descriptions of this review are completely incomprehensible and, therefore, useless.

Reply: Thanks for your comments. Section 2 is in order to understand the strong coupling, such as vacuum Rabi splitting, multiple harmonic oscillators and scattering spectra, and we have improved the theoretical descriptions in this Section in our revised manuscript for a better understanding for the multimode strong coupling.

  1. Further in section 3 the results of experimental studies are described. There are much fewer obvious "inconsistencies" here. However, they are. For example, the question of why the experimental data obtained by different authors can be interpreted as the result of the interaction of excitons and plasmons is not discussed anywhere in this section.

Reply: Thanks for your comments. As we know, many factor will affect the interaction of excitons and plasmons, such as the categories of excitons (MoS2, WS2, J-aggregates,etc.. ) or plasmons (silver nanoprisms,Ag nanodisks, Au nanostars, etc..), experimental temperature, and in our review, we have already discussed in this section 3.

  1. The caption to figure 3 says “Energies of reflectivity dips as a function of the nanodisk diameter extracted from the reflectivity spectrum”. What do the original reflectance spectra look like and how is the data extraction procedure performed?

Reply: Thanks for your comments. And we are sorry to remind reviewer about the caption which says “Energies of reflectivity dips as a function of the nanodisk diameter extracted from the reflectivity spectrum” is figure 4 (b). It depicts resulting dispersion curves of the Ag-WS2 heterostructure that was embedded in the optical microcavity [Opto-Electronic Advances, 2019,2(5):190008]. The original reflectivity spectra of the Ag-WS2 heterostructure with the diameter of Ag nanodisk increased from 85 to 95nm is as shown in supporting Figure S1:

Figure S1. Reflectivity spectra of the Ag-WS2 heterostructure with the diameter of Ag nanodisk increased from 85 to 95nm.

And in figure 4 (b), red dots with error bars show energies obtained from the reflectivity spectrum. The horizontal black dashed lines respectively represent the A-exciton and the microcavity resonant energy. The black slanted short-dashed line represents plasmon resonance mode. Three green solid curves correspond to theoretical fits of hybrid branches based on the three-coupled oscillator model. The error bar represents the standard error of a set of measurements. We have added these in our revised manuscript, which is shown in red on Page 11 in line 421-425.

  1. In Fig. 8 (unfortunately, this is perhaps the only example of a spectrum), only two of the four curves are identified. Are the presented curves experimental or still calculated? If experimental, where is the comparison with the calculated curves?

Reply: Thanks for your good comments of our manuscript. We have identified the four curves in our manuscript, which is shown in red on Page 15 in line 540-543. And the presented curves are experimental, and we have added Figure 8 (b), which is the comparison with the calculated curves in our manuscript, which is shown in red on Page 15 in line 538-547.

  1. In general, section 3 is perceived as some compilation of information from the published works of other researchers, without reasonable assessments, generalizations and logical conclusions.

Reply: Thanks for your good comments of our manuscript. We have reclassified the section into three parts according to the structure style, “Multimode coupling related to microcavity nanostructure”, “Multimode coupling related to periodic noble metallic NPs-J-aggregates nanostructure” and “Multimode coupling related to core-shell multicomponent systems”, respectively, which is with reasonable assessments and generalizations. And this review introduces the theoretical principle of multimode strong coupling based on surface plasmons, which is aim to reveal the physical mechanism of multimode strong coupling, and it is of great significance in the development of nano-optoelectronic devices and quantum communication. Furthermore, the logical conclusions are in the Section 4 in our manuscript.

Reviewer 3 Report

This review is an important step in the analysis of  bibliographic data concerning the couplings between plasmons and excitons. I recommend it for publication. However, I find, as it is a review, that it lacks a few points which seem to me not negligible even if they are not main. :

(1) The contributions of Raman works in the field are too quickly evacuated from my point of view.

(2) In this review, there are clearly no references to the work of the Professor Mlayah team from the CEMES in Toulouse (France): it seems to me that this is a defect! 

(3) No allusion is made either to the possibilities that the nanomanipulations of plasmonizing nanoobjects could open up to design structures allowing or prohibiting plasmon/exciton couplings. (for instance : Materials 2019, 12(9), 1372; doi.org/10.3390/ma12091372) or yet Appl. Sci. 2021, 11(19), 9133; https://doi.org/10.3390/app11199133 or J. Phys. Chem. C 2020, 124, 4, 2705–2711doi.org/10.1021/acs.jpcc.9b10096).

(4) Finally, in the openings or perspectives, could we also open up to aspects that are still little studied in this new physics, what are the potential phase changes that could be induced by these strong couplings? 

Author Response

Reviewer 3

  1. The contributions of Raman works in the field are too quickly evacuated from my point of view.

Reply: Thanks for your suggestion to our manuscript. We have cited the related Raman works in our revised manuscript, which is shown in red on Page 2 in line 64-68.

  1. In this review, there are clearly no references to the work of the Professor Mlayah team from the CEMES in Toulouse (France): it seems to me that this is a defect! 

Reply: Thanks for your suggestion to our manuscript. We have added the work of the Professor Mlayah team from the CEMES in Toulouse (France) (No. 27-30) in our revised manuscript, which is shown in red on Page 2 in line 68-69.

  1. No allusion is made either to the possibilities that the nanomanipulations of plasmonizing nanoobjects could open up to design structures allowing or prohibiting plasmon/exciton couplings. (for instance : Materials 2019, 12(9), 1372; doi.org/10.3390/ma12091372) or yet Appl. Sci. 2021, 11(19), 9133; https://doi.org/10.3390/app11199133 or J. Phys. Chem. C 2020, 124, 4, 2705–2711doi.org/10.1021/acs.jpcc.9b10096).

Reply: Thanks for your good suggestion to our manuscript. We have added the related description in our revised manuscript, which is shown in red on Page 2 in line 97-98.

  1. Finally, in the openings or perspectives, could we also open up to aspects that are still little studied in this new physics, what are the potential phase changes that could be induced by these strong couplings? 

Reply: Thanks for your good suggestion to our manuscript. We have added the related prospect in our revised manuscript, which is shown in red on Page 20 in line 706-711.

Reviewer 4 Report

The authors present a comprehensive review concerning basic principles and applications based on multimode interaction between light and matter.

The paper is suitably organized and well documented. However, this review completely ignores a new direction of interest, concerning machine learning applications in plasmonics. I would suggest the authors add a small section, as perspective applications.

Some typos need to be corrected throughout the manuscript, e.g.:

Line 100:  Dicke Hamiltonnian --> double "n" 
Line 111: formatting errors in Eq. (2)

Author Response

Reviewer4

  1. The authors present a comprehensive review concerning basic principles and applications based on multimode interaction between light and matter. The paper is suitably organized and well documented. However, this review completely ignores a new direction of interest, concerning machine learning applications in plasmonics. I would suggest the authors add a small section, as perspective applications.

Reply: Thanks for your good suggestion to of our manuscript. We have added the related prospect in our revised manuscript, which is shown in red on Page 20 in line 719-725.

  1. Some typos need to be corrected throughout the manuscript, e.g.: Line 100:  Dicke Hamiltonnian --> double "n" ; Line 111: formatting errors in Eq. (2).

Reply: Thanks for your good suggestion to our manuscript. We have corrected the wrong typos in our manuscript, which is shown in red on Page 3 in line 107 and 129, respectively.

Round 2

Reviewer 2 Report

The revised manuscript looks much better than previous version. It can be now published in its present form.